# Denoising Diffusion Path: Attribution Noise Reduction with An Auxiliary Diffusion Model

Yiming Lei[1], Zilong Li[1], Junping Zhang[1], Hongming Shan[2]*
[1] Shanghai Key Laboratory of Intelligent Information Processing,
School of Computer Science, Fudan University
[2] Institute of Science and Technology for Brain-Inspired Intelligence &
MOE Key Laboratory of Computational Neuroscience and Brain-Inspired Intelligence &
MOE Frontiers Center for Brain Science, Fudan University
{ymlei, hmshan}@fudan.edu.cn

## Abstract

The explainability of deep neural networks (DNNs) is critical for trust and reliability in AI systems. Path-based attribution methods, such as integrated gradients (IG), aim to explain predictions by accumulating gradients along a path from a baseline to the target image. However, noise accumulated during this process can significantly distort the explanation. While existing methods primarily concentrate on finding alternative paths to circumvent noise, they overlook a critical issue: intermediate-step images frequently diverge from the distribution of training data, further intensifying the impact of noise. This work presents a novel Denoising Diffusion Path (DDPath) to tackle this challenge by harnessing the power of diffusion models for denoising. By exploiting the inherent ability of diffusion models to progressively remove noise from an image, DDPath constructs a piece-wise linear path. Each segment of this path ensures that samples drawn from a Gaussian distribution are centered around the target image. This approach facilitates a gradual reduction of noise along the path. We further demonstrate that DDPath adheres to essential axiomatic properties for attribution methods and can be seamlessly integrated with existing methods such as IG. Extensive experimental results demonstrate that DDPath can significantly reduce noise in the attributions—resulting in clearer explanations—and achieves better quantitative results than traditional path-based methods.

## 1 Introduction

Deep neural networks (DNNs) have achieved remarkable success in various tasks, but their opaque decision-making processes remain a significant challenge and are critical to those high-staking scenarios like medical diagnosis [1] and autonomous driving [2]. Explainable Artificial Intelligence (XAI) aims to bridge this gap by providing insights into how DNNs make their predictions, where the commonly used interpretation methods include class activation mapping (CAM)-based [3, 4, 5, 6, 7] and path-based methods [8, 9, 10, 11, 12].

Theoretically, the path-based methods comply with the rigorous axiomatic properties, such as the implementation invariance and symmetry-preserving [8], contributing significantly to the interpretation field, and we also focus on this kind of technique. Path-based attribution methods, such as integrated gradients (IG) [8] that is based on game-theoretic idea [13], offer a valuable tool for XAI by accumulating gradients along a path from a baseline image to the target image being explained. However,

---

*Corresponding author.

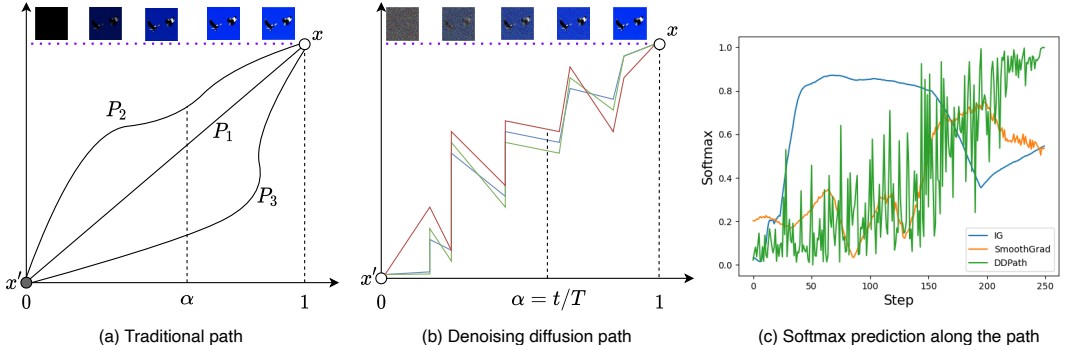

(a) Traditional path      (b) Denoising diffusion path      (c) Softmax prediction along the path

Figure 1: Motivation illustration of DDPath. The symbol $x'$ denotes the baseline image and $x$ the target image. (a) The existing paths are irrelevant to data distributions. (b) The proposed denoising diffusion path approaches the distribution of real data. (c) Traditional IG [8] and SmoothGrad [9] struggle to maintain a continuously increasing Softmax probability along the integration path. This behavior can be counterintuitive and contradict human cognition, where the confidence in a prediction should generally rise as evidence accumulates. In contrast, the proposed DDPath achieves a more natural behavior by ensuring a continuously increasing Softmax probability along the path, even if the path itself exhibits fluctuations.

these methods suffer from a crucial limitation: noise accumulation along the path. This noise can significantly distort the explanation, making it difficult to identify the features truly contributing to the DNN's decision.

Existing approaches primarily focus on finding alternative paths to bypass noise regions. SmoothGrad progressively added noise to the image and achieved the effect of noise reduction, and the authors have verified that adding noise can help reduce noise during inference [9]. Blur IG successively blurred the input image with the Gaussian kernels that varied along the path [10]. Blur IG does not require a pre-defined "baseline" image which is critical to the original IG [8]. Guided IG is a general concept, *i.e.*, a superset of path methods, which avoids the unrelated regions with high gradients by minimizing the attributions at every feature (or pixel) across this superset [11]. While the above alternatives can be beneficial, they neglect an essential issue: during the path construction, the intermediate-step images were modified manually by operations like noising or blurring, *i.e.*, independent of the input image, resulting in them deviating significantly from the data distribution the DNN was trained on. This distribution shift further amplifies noise and hinders interpretability.

In this paper, we intend to reduce explanation noise for path methods from a new perspective: the explanation noise stems from the distribution shift of intermediate-step images when calculating their gradients along the path, because the shifted images offer biased predictions for a pre-trained classification model (Fig. 1(c)), then influence the attributions. Hence, it is necessary to make the distributions of intermediate-step images closer to that of the original input image, so that the gradients back-propagated from accurate predictions are more relative to classes. As shown in Fig. 1(a), the traditional paths $P_1$, $P_2$, and $P_3$ are independent of the input distribution even though they have the same starting point and the endpoint. Therefore, we aim to develop a path that simultaneously approaches the real data and implies progressive noise reduction.

Inspired by the recently advanced diffusion models, which progressively add noise in the forward process and denoising during the reverse sampling process [14, 15, 16], it is natural to *correlate the reverse denoising process with the attribution path*. On the other hand, the reverse process can recover the images that comply with the original data distribution despite the noisy intermediates. To this end, we propose a Denoising Diffusion Path (DDPath) for the attribution of deep neural networks. First, we define a novel denoising diffusion path that aligns the attribution path with the reverse sampling process by scaling the sampling steps. This enables the attribution path to incorporate the ability of generative modeling of diffusion models and the resultant intermediate-step images possess the approximated distributions with that of a classification model. Similar to existing path methods, the DDPath is also approximated by Riemann approximation [8, 12]. Furthermore, we demonstrate that the DDPath satisfies the corresponding axioms. Second, the DDPath can be easily combined

with previous path methods and we developed the DDPath-IG, DDPath-BlurIG, and DDPath-GIG. In practice, we apply the pre-trained classifier-guided diffusion model to construct the DDPath [16]. Note that we do not attempt to investigate many advanced diffusion models in this paper, we pay more attention to exploring the reverse diffusion process to work with DNN attribution, which has not been discussed in previous attribution studies.

**Contributions.** We summarize the main contributions of this paper as follows. (**i**) We propose a novel Denoising Diffusion Path (DDPath) for DNNs attribution. (**ii**) DDPath is theoretically compatible with current path-based methods, and we develop DDPath-IG, DDPath-BlurIG, and DDPath-GIG counterparts enhancing the baseline methods. (**iii**) DDPath can be easily implemented by applying a pre-trained classifier-guided diffusion model. (**iv**) Experimental results demonstrate the effectiveness of DDPath on both qualitative saliency maps and quantitative evaluations of insertion and deletion scores and accuracy information curves (AIC) [17, 12].

## 2   Related Work

**Gradient-based attribution.** Integrated gradients (IG) [8] is designed to address the shortcomings of traditional saliency maps. By integrating gradients along a straight-line path from a baseline image to the target image, IG adheres to the sensitivity axiom and implementation invariance axiom. This property guarantees that the generated explanations are interpretable and consistent. While IG has been a significant advancement in XAI, subsequent research has focused on further improving its performance and addressing its limitations. Boundary-based integrated gradient [18] enhances precision with a boundary search mechanism and better baseline selection, while adversarial gradient integration (AGI) seeks higher accuracy through non-linear ascending trajectories [19]. However, AGI relies heavily on the quality of adversarial samples. Efforts like guided integrated gradients (GIG) address noise in the IG path, but GIG suffers from computational cost and limitations to image data [11]. Similarly, Fast-IG [20] and expected gradient (EG) [21] face limitations in efficiency or dependence on input features. These shortcomings in existing gradient-based methods motivate our work on DDPath. DDPath aims to provide cleaner and more interpretable attributions by tackling noise accumulation along the integration path. DDPath is designed to realize the progressive emergence of the image signal and gradual noise reduction along the path.

**Classifier-guided diffusion models.** Recent advanced diffusion models like score-based diffusion models [22] and denoising diffusion probabilistic models (DDPMs) [14] have greatly facilitated the progress of generative modeling tasks. Of particular relevance to our work is the concept of classifier-guided diffusion models introduced by Ho *et al.* [23]. This framework guides the diffusion process using a pre-trained classifier, essentially learning to "reverse" the noise addition and correctly reach an input the classifier recognizes. This establishes a crucial link between diffusion models and classification tasks, paving the way for their application in interpretability, which is precisely the focus of DDPath. By leveraging this concept, DDPath benefits from the model's ability to progressively remove noise. This denoising capability directly tackles the challenge of noise accumulation in the attribution path, leading to cleaner and more interpretable explanations. Notably, explicitly constructing an attribution path with diffusion models has not been discussed in current attribution studies.

## 3   Preliminary

Before introducing our DDPath, we recall the path-based attribution framework and the corresponding axioms to be satisfied.

Integrated gradients (IG) [8] is a pioneer work for deep visual model attribution with complete axiomatic properties, resulting in enormous axiomatic-based attribution methods. Assume that the $F$ is a deep neural network to be explained, IG accumulates gradients along a linear path $\gamma^{\mathrm{IG}}(\alpha) = \boldsymbol{x}' + \alpha(\boldsymbol{x} - \boldsymbol{x}')$ via path integration:

$$\mathcal{A}_i = \int_0^1 \frac{\partial F(\gamma^{\mathrm{IG}}(\alpha))}{\partial \gamma_i^{\mathrm{IG}}(\alpha)} \frac{\partial \gamma_i^{\mathrm{IG}}(\alpha)}{\partial \alpha} \, \mathrm{d}\alpha. \tag{1}$$

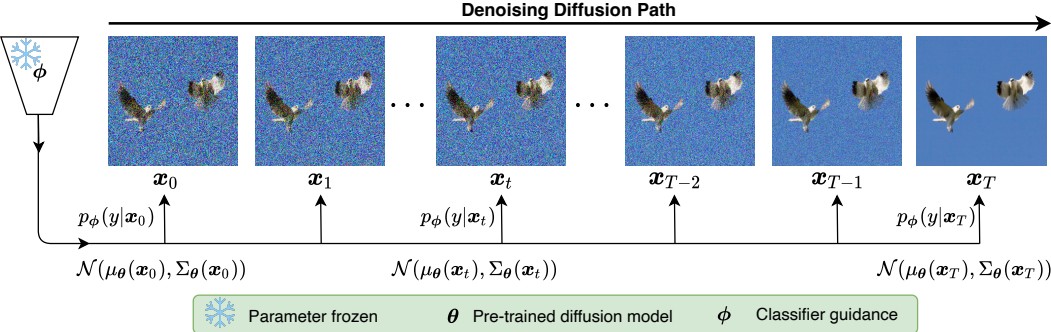

Figure 2: Illustration of DDPath. At each step in the DDPath, the images are sampled from a pre-trained diffusion model $\theta$ guided by a classifier $\phi$.

Current approaches focus more on finding a better path, *i.e.*, $\frac{\partial \gamma(\alpha)}{\partial \alpha}$, ignoring the intermediate points $\frac{\partial f(\gamma(\alpha))}{\partial \gamma_i(\alpha)}$ that is referred to as the inherent distribution shifts. On the other hand, an appropriate baseline is essential to traditional path-based attribution methods. The black baseline (or black image) suits IG better than a noise baseline, and this also causes difficulty in attributing black or dark regions of interest. Andrei *et al.* addressed this obstacle by applying both black and white baselines [17]. Hence, these manually designed baselines have their own biases. In this paper, the proposed DDPath can simply work with a noise baseline with which existing methods cannot work well.

**Sensitivity.** The sensitivity axiom dictates that if a single feature changes between a baseline and a target image while causing different predictions, the attribution method must assign a non-zero attribution score to the differing feature. Otherwise, the attribution method might be insensitive to crucial changes.

**Implementation invariance.** An attribution method follows the implementation invariance principle when, for the same pair of input data and predicted output, regardless of the specific neural network architecture or implementation details, the attribution scores remain consistent.

## 4 Denoising Diffusion Path

For the target image $x$ and a baseline $x'$, traditional attribution mainly considers $x_i - x'_i$, *i.e.*, the differences between the $i$-th feature of the image and baseline, measuring how the classification model can behave with the gradual appearance of the $i$-th feature. In our diffusion path, such differences turned to *the gradual appearance of images while the disappearance of noises*, there are no direct relationships between intermediate steps and the original noisy baseline in that the sampling of the reverse diffusion process generates these intermediate images. It is practically implemented with Riemann approximation that will be discussed in Sec. 4.3.

**Definition 1.** *(Denoising Diffusion Path or DDPath) This path is a piece-wise linear function [24, 25] built upon the reversely sampled sequence with a maximum step number $T$: $x_T, \ldots, x_t, x_{t-1}, \ldots, x_0$, which is defined as*

$$\gamma(\alpha) = \alpha x_\alpha, \quad x_\alpha \sim \mathcal{N}_\alpha(\mu_\alpha(x), \Sigma_\alpha(x)), \tag{2}$$

*where $x_T$ is the noisy signal baseline and $x_0$ is the finally sampled image, each piece $\mathcal{N}_\alpha(\mu_\alpha(x), \Sigma_\alpha(x))$ is a set of samples complied with a Gaussian distribution with mean and variance concerning the target image $x$. The diffusion sample step $t$ is aligned with the path coefficient $\alpha$ by setting $\alpha = \frac{t}{T} \in [0, 1]$.*

Theoretically, DDPath is also a type of definition in terms of a set of all possible paths, which is similar to the adaptive path in GuidedIG [11] and Shapley values [26]. That is to say, for each loop from the baseline to the target image, the path is composed of sampled images from every piece $\mathcal{N}_\alpha(\mu_\alpha(x), \Sigma_\alpha(x))$. For the rest of this paper, the $\gamma(\alpha)$ denotes the proposed DDPath if without a specific statement.

### 4.1 Attribution with DDPath

Based on Definition 1, we discuss how to attribute along the DDPath. Specifically, we showcase that DDPath-IG, DDPath-BlurIG, and DDPath-GuidedIG enhance the corresponding baseline methods. First, for the DDPath-IG, we can directly replace the linear path in Eq. (1) with the DDPath:

**Definition 2.** *(DDPath-IG) Given a diffusion model $\mathcal{E}_\theta$ pre-trained using classifier guidance, $f_\phi$ is the corresponding pre-trained classifier, for the $i$-th feature in the input $\boldsymbol{x}$ of class $y$, its attribution is the integrated gradients along the DDPath:*

$$DDPath\text{-}IG \triangleq \mathcal{A}_i = \int_{\alpha=0}^{1} \frac{\partial F(\gamma(\alpha))}{\partial \gamma_i(\alpha)} \frac{\partial \gamma_i(\alpha)}{\partial \alpha} \, \mathrm{d}\alpha, \ \gamma(\alpha) = \alpha \boldsymbol{x}_\alpha, \ \boldsymbol{x}_\alpha \sim \mathcal{N}_\alpha(\hat{\mu}_\theta(\boldsymbol{x}), \Sigma_\theta(\boldsymbol{x})), \quad (3)$$

*the $\mathcal{N}_\alpha(\hat{\mu}_\theta(\boldsymbol{x}), \Sigma_\theta(\boldsymbol{x}))$ denotes the distribution parameterized by the diffusion model $\mathcal{E}_\theta$, and $\hat{\mu}_\theta(\boldsymbol{x}) = \rho \cdot \mu_\theta(\boldsymbol{x}) + \kappa \cdot \Sigma \nabla_{\boldsymbol{x}_\alpha} \log p_\phi(y|\boldsymbol{x}_\alpha)$. The $\rho$ and $\kappa$ are scaling factors controlling the mean and the gradient term variation.*

A critical problem of sampling with the diffusion model is that the generated images are diverse when sampling from the noise signals. In DDPath-IG, we solve this obstacle by simply enforcing the sampling centered at the target image, *i.e.*, the mean and variance are calculated by the original image $\boldsymbol{x}$. We use the same sampling strategy for the following DDPath-BlurIG and DDPath-GIG.

**Definition 3.** *(DDPath-BlurIG) Given the Gaussian kernels along the path parameter $\alpha$, $L(x, y, \alpha) = \sum_{m=-\infty}^{\infty} \sum_{n=-\infty}^{\infty} \frac{1}{\pi\alpha} e^{-\frac{x^2+y^2}{\alpha}} \cdot \gamma(\alpha)(x-m, y-n)$, then the attribution of the $i$-th feature is obtained by:*

$$DDPath\text{-}BlurIG \triangleq \mathcal{A}_i = \sum_{t=1}^{T} \frac{\partial F(L(x, y, \alpha))}{\partial L(x, y, \alpha_t)} \frac{\partial L(x, y, \alpha_t)}{\partial \alpha_t} \frac{\alpha_t}{T}, \qquad (4)$$

*where the $t$ is the number of steps in the Riemann approximation, and $\alpha_t = t \cdot \frac{\alpha_t}{T}$ [10].*

DDPath-BlurIG applies the DDPath and blurs the sampled images along the denoising path. That is to say, it scales the spaces of all sampled pieces in Definition 3 while preserving the data distributions within pieces to approach the real data distribution in Fig. 1.

**Definition 4.** *(DDPath-GIG) Given an IG path $\gamma^{IG}(\alpha)$ [8] and a DDPath $\gamma(\alpha)$, the objective of DDPath-GIG is defined as:*

$$DDPath\text{-}GIG \triangleq \arg\min_{\gamma \in \Gamma} \sum_{i=1}^{N} \int_{\alpha=0}^{1} \frac{\partial F(\gamma(\alpha))}{\partial \gamma_i(\alpha)} \frac{\partial \gamma_i(\alpha)}{\partial \alpha} \, \mathrm{d}\alpha + \lambda \int_{\alpha=0}^{1} \|\gamma(\alpha) - \gamma^{IG(\alpha)}\| \, \mathrm{d}\alpha, \quad (5)$$

*where $\lambda$ is the coefficient that balances the two terms, $N$ is the number of features (or pixels), and $\Gamma$ contains all possible paths of DDPath.*

In Definition 4, the traditional path is replaced with our DDPath, and in practice, the $\Gamma$ is implemented by repeating random sampling loops. Therefore, the first term of Eq. (5) aims to find a better denoising path $\gamma(\alpha)$ that avoids those regions causing noisy explanations, and the second term ensures the diffusion path does not deviate severely off the shortest path, decreasing the likelihood of crossing areas that are too out-of-distribution.

### 4.2 Axiomatic Properties of DDPath-IG

In this section, taking DDPath-IG as an example, we show that it satisfies the axiomatic properties in [8]. First, the DDPath-IG satisfies the *sensitivity* that the image differs from the noisy baseline, and then the partial derivatives of differing features are non-zero. Second, the DDPath is agnostic to the architecture of DNNs so that the DDPath-IG also satisfies the *completeness* that $\sum_i^N \mathcal{A}_i = F(\boldsymbol{x}) - F(\boldsymbol{x}')$ [27, 28, 8]. Third, the calculation of partial derivatives of DDPath-IG follows the chain rule so that the attributions are invariant to network implementations, therefore the DDPath-IG satisfies the *implementation invariance*. Fourth, recall that the IG is the unique path method that is *symmetry-preserving* (Theorem 1 in [8]), the DDPath-IG also maintains this property.

**Algorithm 1** Algorithm of DDPath-IG.

---

**Require:** Target image $\boldsymbol{x}$ and its label $y$, the initial noisy baseline $\boldsymbol{x}'$ randomly sampled from a Gaussian; target model $F(\cdot)$, diffusion trained classifier $h_\phi$, the diffusion model $\mathcal{E}_\theta$, total number of step $T$.

**Return:** Attribution for the target image $\boldsymbol{x}$: $\mathcal{A} = \frac{1}{T} \sum_{t=0}^{T-1} g_t$.

1: **for** $t = 0$ to $T - 1$ **do**
2:      **if** t == 0 **then**
3:          $\boldsymbol{x}_t = \boldsymbol{x}'$                                               ▷ Noise baseline
4:      **else**
5:          $\boldsymbol{x}_t = \boldsymbol{x}'_t$                                   ▷ Sampled image in $t - 1$ step
6:      **end if**
7:      $\rho = 1 - \frac{t}{T}, \kappa = \frac{t}{T}$                         ▷ Scaling coefficients
8:      $\hat{\mu}_\theta(\boldsymbol{x}) = \rho \cdot \mu_\theta(\boldsymbol{x}) + \kappa \cdot \Sigma \nabla_{\boldsymbol{x}_t} \log p_\phi(y | \boldsymbol{x}_t)$.     ▷ Update sampling mean
9:      $\boldsymbol{x}'_t \sim \mathcal{N}_t(\hat{\mu}_\theta(\boldsymbol{x}), \Sigma_\theta(\boldsymbol{x}))$.                           ▷ Sampling
10:     $g_t = \frac{\partial(F(\boldsymbol{x}_t))}{\partial \boldsymbol{x}_t}$.                                  ▷ Calculate gradients
11: **end for**

---

## 4.3 Implementation with Classifier-Guided Diffusion Sampling

To ensure clarity, we reiterate that this paper concentrates on establishing a correlation between the attribution path and the reverse diffusion process. Specifically, we implement the DDPath-IG with a pre-trained diffusion model trained with classifier guidance [16]. The algorithm of DDPath-IG is shown in Algorithm 1, and the algorithms of DDPath-BlurIG and DDPath-GIG are provided in the Appendix. The implementation of the integral is also approximated by the Riemman approximation, taking DDPath-IG as an example:

$$\mathcal{A} = \lim_{T \to \infty} \sum_{t=1}^{T} \frac{\partial F(\boldsymbol{x}_t)}{\partial \boldsymbol{x}_t} \cdot \frac{t}{T} \boldsymbol{x}_t. \tag{6}$$

For the classifier-guided diffusion sampling in [16], the sampling mean $\hat{\mu}_\theta(\boldsymbol{x}) = \mu + s \Sigma \nabla_{\boldsymbol{x}_t} \log p_\phi(y | \boldsymbol{x}_t)$ scales the classifier gradient term with $s$ that corresponds to the sampling steps $t$. However, in this work, the proposed denoising path should guarantee consistency between the baseline and the target image. Therefore, in Definition 2, the sampling mean is centered at the image $\boldsymbol{x}$ while the classifier gradients are calculated via step-wise intermediate images $\boldsymbol{x}_t$, and we use a simple scaling scheme with $\rho$ and $\kappa$:

$$\rho = 1 - \frac{t}{T}, \quad \kappa = \frac{t}{T}. \tag{7}$$

This scheme is reasonable in that at the very beginning of sampling, we do not expect the mean to shift severely ensuring less data distribution shift. For the variance term, the gradients $\nabla_{\boldsymbol{x}_t}$ are noisy because of the inaccurate predictions at initial steps. With the increase of $t$, more accurate gradients can be obtained by more correct predictions. Hence, the sampling means $\hat{\mu}_\theta(\boldsymbol{x})$ become mainly dominated by this gradient of step-wise inputs, demonstrating that the DDPath progressively enables the increasing prediction scores by preserving data distribution along the path.

## 5 Experiments

### 5.1 Experimental Setup

**Datasets.** Following previous studies [10, 11, 12], we evaluate the effectiveness of DDPath on the validation set of ImageNet-1k [30] that contains $50,000$ images of $1,000$ classes. Furthermore, we conducted a pointing game experiment on MS COCO validation set [31].

**Models and baselines.** For the classification model, we use ResNet-50 [32] and VGG-16/-19 [29] as backbone. The baseline attribution methods are Guided-BP [33], IG [8], Smooth IG [9], Blur IG [10], and Guided IG [11]. All the experiments are implemented by PyTorch [34] and conducted on an NVIDIA A100 GPU. The number of sampling steps for DDPath methods is 250.

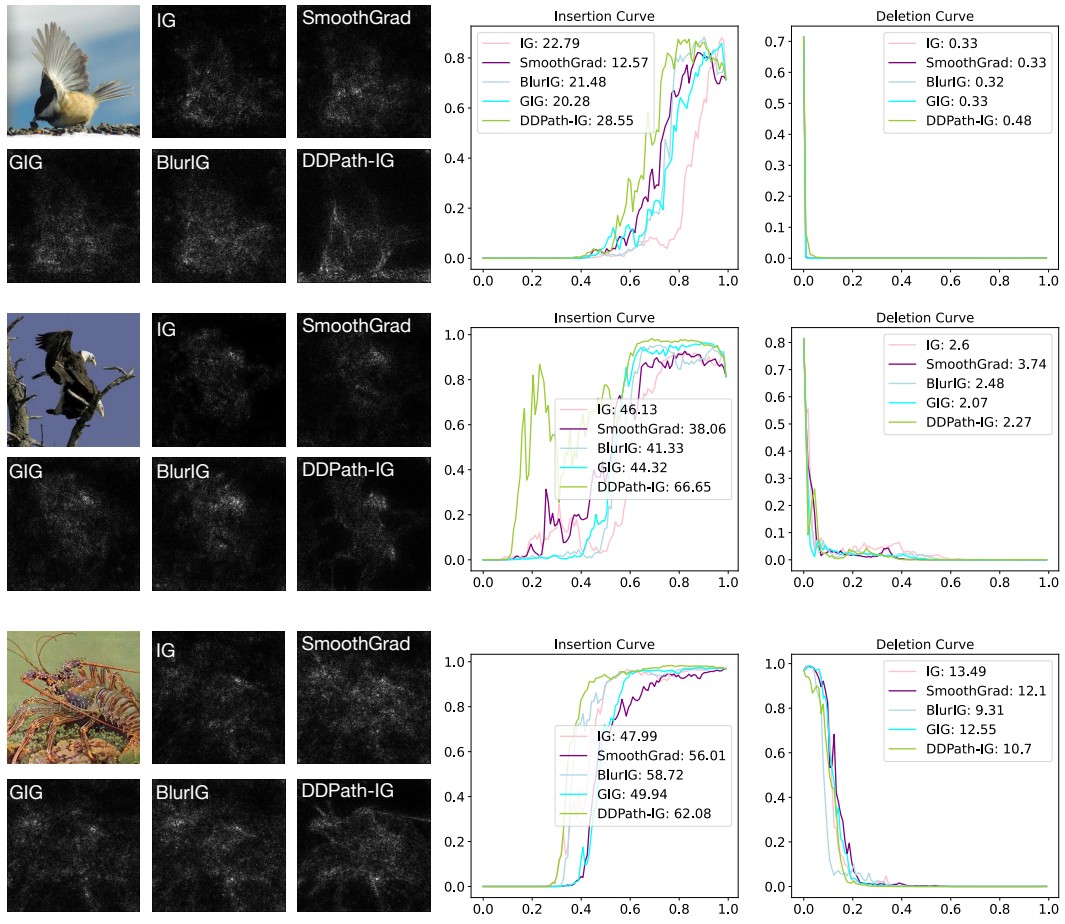

Figure 3: Comparison of saliency maps and corresponding Insertion and Deletion curves. Image examples are selected from the ImageNet-1k validation set. The classification model is the pre-trained VGG-19 [29].

## 5.2 Interpretation Ability

Following previous DNN interpretation works [35, 6, 7, 36], we compare the Insertion and Deletion scores and provide the corresponding curves and the saliency maps in Fig. 3. The insertion process starts with a blurred image, and then iteratively injects original image information (3.6% of total pixels) into the blurred version, guided by the saliency map values. Regions with higher saliency scores are prioritized for insertion, gradually revealing the informative parts of the original image and leading to its full reconstruction. Conversely, deletion identifies relevant pixels (3.6%) in the blurred image based on the saliency map and replaces them with their corresponding values from the original image. This process essentially "unmasks" the informative regions by strategically replacing noise with the original content. In Fig. 3, we can see that the DDPath-IG captures more comprehensive information and more details of the object, *e.g.*, the birds' wings and lobster feet. For the insertion and deletion curves, the DDPath achieves better quantitative insertion AUC values, indicating the image information progressively injected is more important. Although DDPath does not obtain the best deletion AUCs, this is trivial to significant improvement on Insertion and better saliency maps. More saliency maps are provided in Figs. 6 and 7 in Appendix.

## 5.3 Length of Path

In this section, we investigate the path length (or sampling steps for DDPath). Fig. 4 shows the saliency maps of different methods at increased steps. With the path length increase, baseline

Table 1: Quantitative comparisons of different interpretation methods on ImageNet validation set in terms of Insertion and Deletion. Overall = Insertion - Deletion.

| Model | Metric | Guided BP | IG | Smooth Grad | BlurIG | DDPath -BlurIG | GIG | DDPath -GIG | DDPath -IG |
|---|---|---|---|---|---|---|---|---|---|
| VGG-16 | Insertion↑ | 21.2 | 21.7 | 20.9 | 19.2 | 22.3 | 21.2 | 23.5 | **25.9** |
| | Deletion↓ | 15.4 | 14.8 | 15.0 | 13.3 | 13.5 | 14.1 | 13.8 | **12.7** |
| | Overall↑ | 5.8 | 6.9 | 5.9 | 5.9 | 8.8 | 7.1 | 9.7 | **13.2** |
| VGG-19 | Insertion↑ | 21.8 | 23.2 | 21.1 | 20.6 | 24.1 | 22.4 | 25.6 | **27.8** |
| | Deletion↓ | 14.0 | 13.5 | 13.8 | 13.2 | 14.0 | 12.4 | 12.3 | **12.1** |
| | Overall↑ | 7.8 | 9.7 | 7.3 | 7.4 | 10.1 | 10.0 | 13.3 | **15.7** |
| ResNet-50 | Insertion↑ | 32.2 | 33.8 | 32.5 | 25.6 | 27.8 | 36.4 | 38.9 | **45.1** |
| | Deletion↓ | 13.8 | 13.5 | 13.2 | 12.8 | 12.4 | 12.5 | **12.0** | 12.7 |
| | Overall↑ | 18.4 | 20.3 | 18.3 | 12.8 | 15.4 | 23.9 | 26.9 | **32.4** |

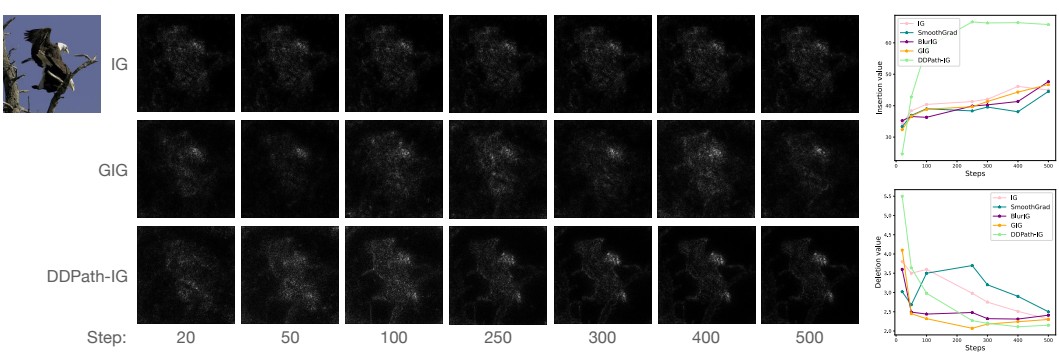

Figure 4: Comparison of saliency maps and corresponding Insertion and Deletion curves obtained by different methods. Image examples are selected from the ImageNet-1k validation set. The classification model is the pre-trained VGG-19 [29].

methods are difficult to obtain interpretation performance gains further while achieving attribution noise reduction. Although they achieve considerable saliency maps at early steps, they still exhibit a weaker noise reduction effect. IG exhibits little variances on saliency maps which is caused by linear path. GIG tends to find an adaptive path resulting in less structural consistency along the whole path. The DDPath-IG provides us with a consistent emergence of the salient regions while preserving consistent structures of the object. From the curves at the right part in Fig. 4, we can see that DDPath performs worse at the early stages, and it obtains more improvements by increasing the path length, however, the baselines gain less.

## 5.4 Pointing Game on COCO

To evaluate the effectiveness of DDPath in pinpointing the most salient pixels, we conducted a "pointing game" on the MS COCO 2017 validation set. This approach, similar to those used in Score-CAM [6] and Group-CAM [7], assesses localization accuracy. We calculated the metric $\frac{\text{Hits}}{\text{Hits}+\text{Misses}}$ to quantify how well the identified salient pixels coincided with the annotated bounding boxes in the data, where the "Hits" counts the number of the most salient pixels that fall in the bounding box, and "Misses" otherwise. Higher scores indicate that DDPath excels at highlighting the most relevant image regions for the model's prediction. In Table 2, the DDPath counterparts consistently outperform the baseline methods, and we argue that this is severely caused by the noisy salient pixels shifted away from the target objects, *i.e.*, out of the bounding box. For example, in Fig. 3, the GIG of the first case, GIG and BlurIG in the second case, and SmoothGrad in the third case. Therefore, the DDPath not only reduces those noises but also avoids the incorrect salient pixels.

Table 2: Pointing game evaluation on MS COCO 2017 validation set.

| Model | IG | DDPath-IG | BlurIG | DDPath-BlurIG | GIG | DDPath-GIG |
|---|---|---|---|---|---|---|
| VGG-16 | 42.3 | 44.7 | 45.0 | 47.2 | 45.2 | 46.3 |
| VGG-19 | 43.4 | 45.2 | 45.3 | 48.9 | 44.9 | 47.1 |
| ResNet-50 | 45.2 | 46.9 | 46.6 | 50.0 | 47.2 | 50.5 |

## 5.5 Ablation Study

**Scaling scheme.** First, we discuss the scaling scheme defined in Eq. (7). The above experiments involving DDPath used $\rho = 1 - \frac{t}{T}$, $\kappa = \frac{t}{T}$ by default, which maintains a stably decreased weight for sampling mean. Here, we reverse such scaling as $\rho = \frac{t}{T}$, $\kappa = 1 - \frac{t}{T}$ to enable a smaller mean and larger variance at the initial steps.

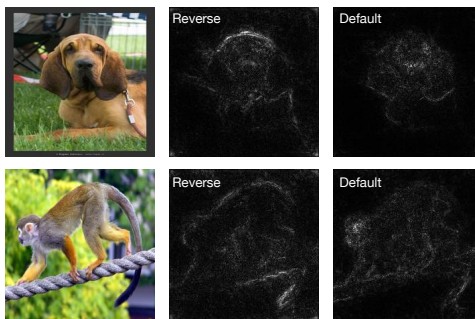

In Table 3, we compare the two scaling schemes in terms of Insertion (Ins.), Deletion (Del.), and accuracy information curves (AIC) [17, 11]. We can see that the Reverse setting performs worse than the Default, demonstrating that sampling with a smaller mean and a larger variance at early steps is inferior in preserving information from the input image. Compared with the baseline methods, the DDPath with Reverse scaling performs better in AIC and Insertion, showing the considerable effect of avoiding distribution shift with DDPath. However, DDPath with Reverse scaling achieves higher deletion values, and we claim that this is due to the larger initial variance caused edge detection effect over the whole image, which compromises the target object. In Fig. 5, the Reverse results highlight more noises and more edges, neglecting the inner regions of the objects.

Figure 5: Saliency maps by different scaling schemes.

**Attribution with noise.** Here, we compare the manually adding noise with our DDPath. In Table 4, we implement Noise counterparts for IG, BlurIG, and GIG. SmoothGrad is also a method of constructing a noisy path. Compared with Table 1, IG-Noise, BlurIG-Noise, and GIG-Noise obtain slight improvements against the vanilla versions in terms of AIC and Insertion, but they are still inferior to DDPath versions. This verifies that the DDPath enables better noise reduction and accurate predictions for the points on the path.

Table 3: Ablation on Scaling Scheme.

| Method | Scaling | AIC↑ | Ins.↑ | Del.↓ |
|---|---|---|---|---|
| IG | - | 15.3 | 23.3 | 13.5 |
| DDPath-IG | Default | **18.9** | **27.8** | **12.1** |
|  | Reverse | 16.2 | 24.8 | 14.5 |
| BlurIG | - | 20.4 | 20.6 | **13.2** |
| DDPath-BlurIG | Default | **24.5** | **24.1** | 14.0 |
|  | Reverse | 20.2 | 21.9 | 14.3 |
| GIG | - | 15.0 | 22.4 | 12.4 |
| DDPath-GIG | Default | **19.7** | **25.6** | **12.3** |
|  | Reverse | 17.7 | 24.5 | 13.2 |

Table 4: Comparison of Adding Noise.

| Method | AIC↑ | Ins.↑ | Del.↓ |
|---|---|---|---|
| IG-Noise | 15.8 | **23.5** | 13.3 |
| BlurIG-Noise | **21.5** | 20.6 | 13.6 |
| GIG-Noise | 14.3 | 21.0 | **12.5** |
| SmoothGrad | 16.6 | 21.1 | 13.8 |

## 6 Conclusion

This paper introduces the Denoising Diffusion Path (DDPath), a novel approach for mitigating noise accumulation in path-based attribution methods. DDPath leverages the power of diffusion models to construct a path where noise is progressively removed, leading to significantly cleaner and more interpretable attributions. We demonstrate that DDPath adheres to essential axiomatic properties and integrates seamlessly with existing methods like Integrated Gradients, requiring only

a pre-trained classifier-guided diffusion model. Extensive evaluations showcase the superiority of DDPath compared to traditional path-based methods, achieving explanations with less noise and better alignment with the DNN's decision-making process.

**Broader impact and limitation.** This paper brought new insights into the attribution of DNNs with simple implementations, *i.e.*, classifier-guided diffusion, and it will trigger more related research in this direction by applying more advanced diffusion models. Moreover, it also provides a possible way of investigating the knowledge of large language models via diffusion models to realize the true human-understandable vision-language consistent explanations. A key limitation of DDPath is that it requires a longer path (more sampling steps) than current methods.

## Acknowledgements

This work was supported in part by National Natural Science Foundation of China (Nos. 62306075, 62101136, 62471148, and 62176059), and China Postdoctoral Science Foundation (No. 2022TQ0069).

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

# A  Appendix

## A.1  Algorithms of DDPath-BlurIG and DDPath-GIG

---

**Algorithm 2** Algorithm of DDPath-BlurIG.

---

**Require:** Target image $x$ and its label $y$, the initial noisy baseline $x'$ randomly sampled from a Gaussian; target model $F(\cdot)$, diffusion trained classifier $h_\phi$, the diffusion model $\mathcal{E}_\theta$, total number of step $T$, the $t$-step Gaussian blur kernels $L(t)$.

**Return:** Attribution for the target image $x$, $\mathcal{A} = \frac{1}{T}\sum_{t=0}^{T-1} g_t$.

  1: **for** $t = 0$ to $T-1$ **do**

  2:     **if** t == 0 **then**

  3:        $x_t = x'$                                        ▷ Noise baseline

  4:     **else**

  5:        $x_t = x'_t$                                      ▷ Sampled image in $t-1$ step

  6:     **end if**

  7:     $\rho = 1 - \frac{t}{T}, \kappa = \frac{t}{T}$                                ▷ Scaling coefficients

  8:     $\hat{\mu}_\theta(x) = \rho \cdot \mu_\theta(x) + \kappa \cdot \Sigma \nabla_{x_t} \log p_\phi(y|x_t)$.      ▷ Update sampling mean

  9:     $x'_t \sim \mathcal{N}_t(\hat{\mu}_\theta(x), \Sigma_\theta(x)); x'_t = L(x'_t, t)$    ▷ Sampling and Gaussian blur on sampled image

10:     $g_t = \frac{\partial(F(x_t))}{\partial x_t}$.                                      ▷ Calculate gradients

11: **end for**

---

---

**Algorithm 3** Algorithm of DDPath-GIG.

---

**Require:** Target image $X^I$, noise baseline $X^B$ randomly sampled from Gaussian, target model $F(\cdot)$, diffusion trained classifier $h_\phi$, the diffusion model $\mathcal{E}_\theta$, total number of step $T$, gradient of the function $grad(x)$, target fraction of features to change at each step $p \in (0, 1]$.

**Return:** $attr$, the attribution for target image $X^I$.

  1: $d_{total} \leftarrow ||X^B - X^I||_1, x \leftarrow X^B, attr \leftarrow zeros(\text{size of} X^I)$      ▷ Initilization

  2: **for** $t \leftarrow 1$ to $T$ **do**

  3:     **if** t == 0 **then**

  4:        $x = X^B$                                        ▷ Assign noise baseline

  5:     **else**

  6:        $x = x'_t$                                        ▷ Sampled image in $t-1$ step

  7:     **end if**

  8:     $\rho = 1 - \frac{t}{T}, \kappa = \frac{t}{T}$                                ▷ Scaling coefficients

  9:     $\hat{\mu}_\theta(x) = \rho \cdot \mu_\theta(x) + \kappa \cdot \Sigma \nabla_{x_t} \log p_\phi(y|x_t)$.      ▷ Update sampling mean

10:     $x \leftarrow x'_t \sim \mathcal{N}_t(\hat{\mu}_\theta(x), \Sigma_\theta(x))$.                          ▷ Sampling

11:     **repeat until** $\delta \leq 1$

12:     $y_i \leftarrow \infty, \forall I | x_i = X_i^I; d_{target} \leftarrow d_{total}(1 - \frac{t}{T}); d_{current} \leftarrow ||x - X^I||_1$

13:

14:     **if** $d_{target} = d_{current}$ **then**

15:        **break**

16:     **end if**

17:     Assign to $S$ the $p$ fraction of features with the lowest absolute gradient values:

18:     $S \leftarrow i | |y_i| \leq fraction(p, |y|)$

19:     $d_S \leftarrow \sum_{i \in S} |x_i - X_i^I|; \delta \leftarrow \frac{d_{current} - d_{target}}{d_S}; \text{temp} \leftarrow x$

20:     **if** $\delta > 1$ **then**

21:        $x_i \leftarrow X_i^I, \forall \in S$

22:     **else**

23:        $x_i \leftarrow (1 - \delta)x_i + \delta X_i^I, \forall i \in S$

24:     **end if**

25:     $y_i = 0, \forall i = \infty; attr_i = attr_i + (x_i - temp_i)y_i, \forall i \in S$

26: **end for**

---

## A.2 More results of saliency maps obtained by VGG-19

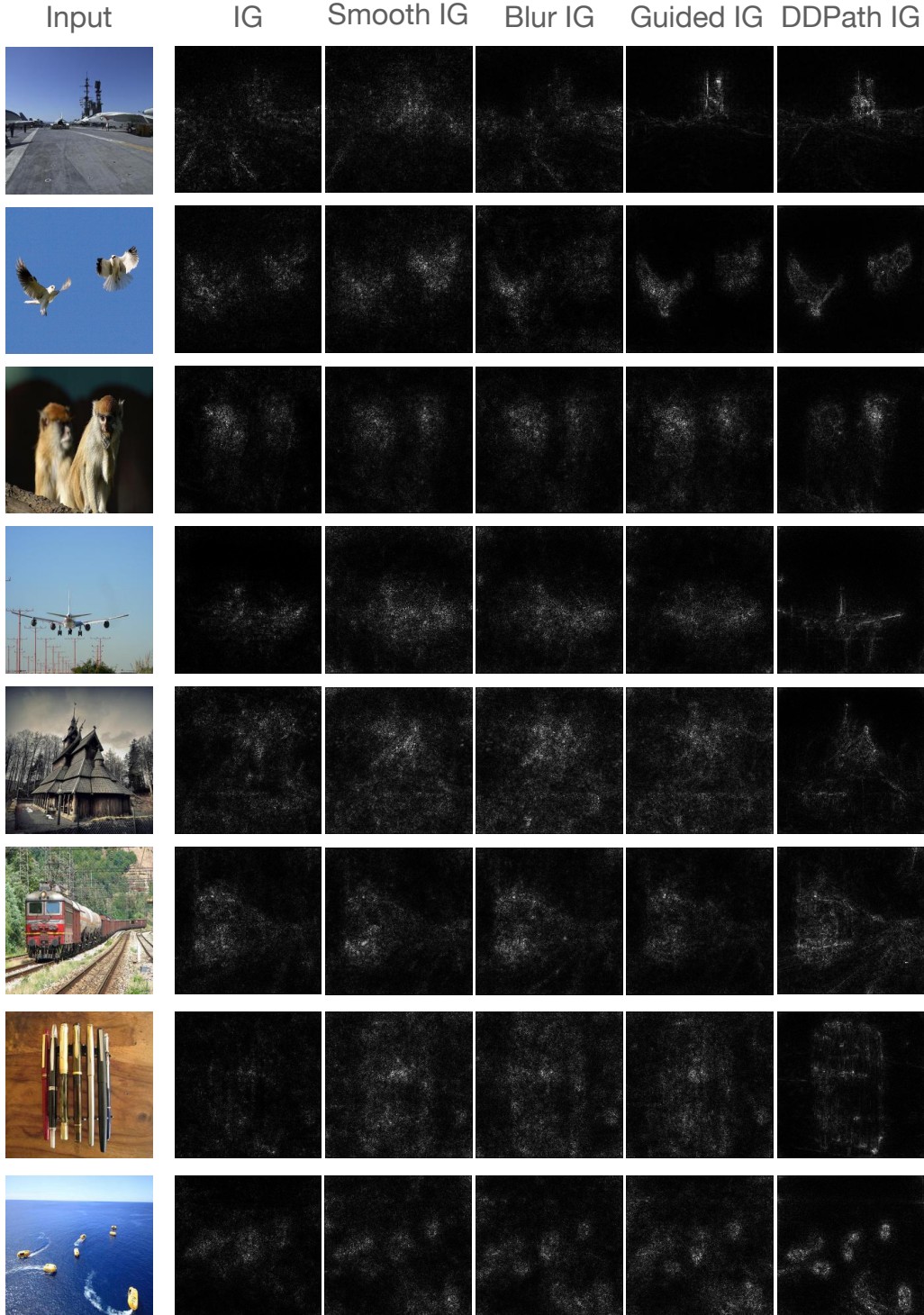

Figure 6: Saliency maps obtained by VGG-19.

## A.3 More results of saliency maps obtained by ResNet-50

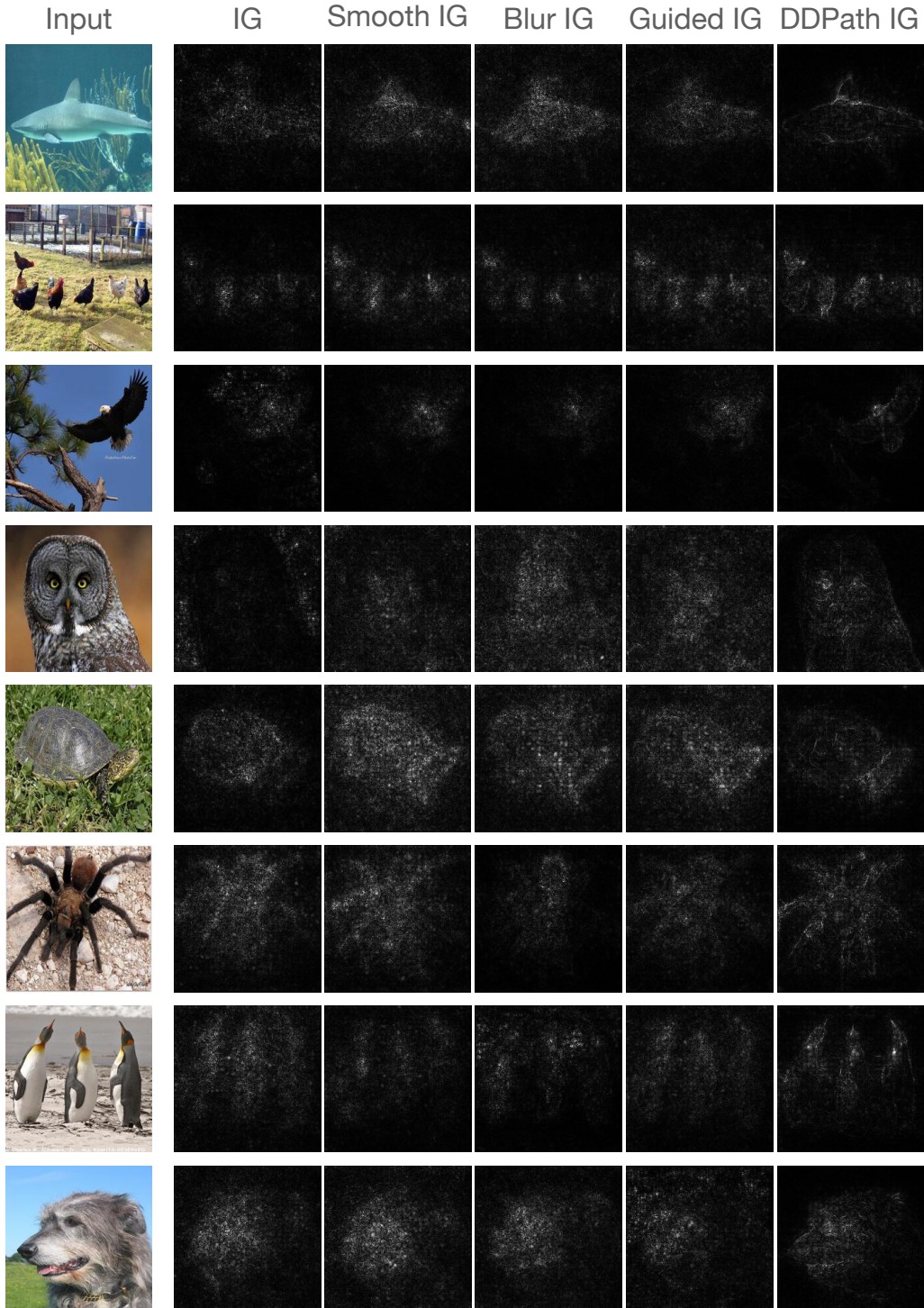

Figure 7: Saliency maps obtained by ResNet-50.

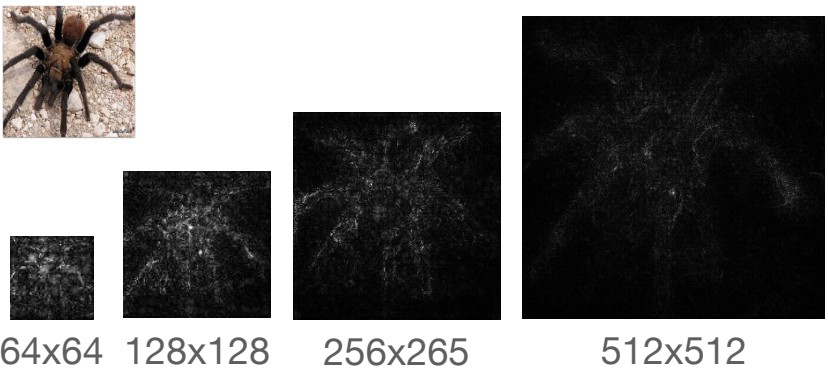

64x64   128x128   256x265   512x512

Figure 8: Saliency maps generated by DDPath-IG using diffusion models of varying sizes.

### A.4 Different diffusion model sizes

To investigate the diffusion model size, we apply the released diffusion models by [16]. Note that these diffusion models of different sizes correspond to different image resolutions, including $64 \times 64$, $128 \times 128$, $256 \times 256$, and $512 \times 512$. The visualization results can be found in Fig. 8. We can see that larger models generate larger resolutions of saliency maps, and they illustrate more fine-grained details.

### A.5 Effects on adversarial examples

We applied two approaches to generate adversarial samples, one is the fast gradient sign attack (FGSM) described by Goodfellow *et al.* [37], and the other is adding simple Gaussian noise. We compared the results of IG and DDPath-IG in terms of Insertion and Deletion values, these results and the saliency maps are shown in Fig. 9. Interestingly, the IG generated saliency maps with degraded quality, while the DDPath-IG are more robust to adversarial samples (FGSM and Gaussian).

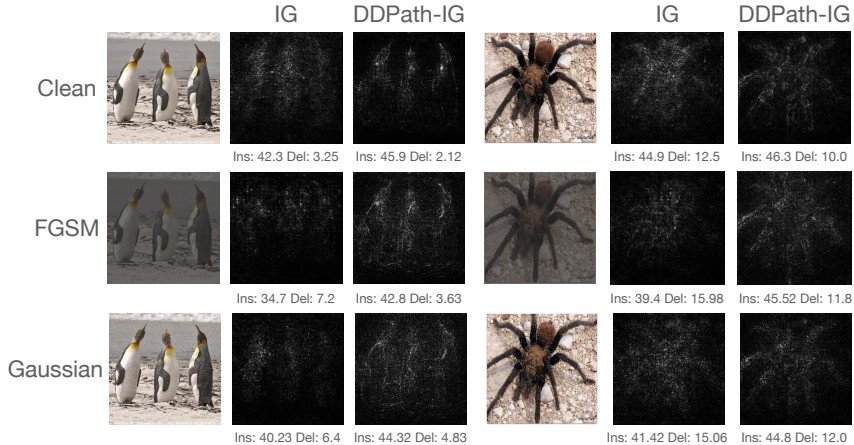

Figure 9: Saliency maps for adversarial examples generated by FGSM and Gaussian.

### A.6 Additional scaling scheme

We evaluated our DDPath by setting $\rho = 1 - (\frac{t}{T})^a$ and $\kappa = (\frac{t}{T})^a$ with both $a = 0.5$ and $a = 2$, and we note that the linear scaling used in this paper is equal to that of $a = 1$. As shown in Table 5, we can see that the DDPath-IG surpasses the baseline IG among different $a$ values, indicating the effectiveness of our DDPath. When $a = 2$, the weight of the mean term decreases slowly at the early step, ensuring better preservation of the main object in the images. Besides, the weight of the class-related variance term increases fast at higher steps, enabling better preservation of discriminative information and object details, and this is consistent with the mechanism of task weights in [38]. In

contrast, when $a = 0.5$, the variance weight increases fast at early steps while the noises are still severe. Hence, the class-related information can be affected by the noises while influencing the classification results and attribution qualities as shown in Fig. 10. The setting of $a = 1$ is a trade-off in our experiments.

Table 5: Scaling with different $a$ values using VGG-19 target model.

| Model | IG | DDPath-IG (0.5) | DDPath-IG (1.0) | DDPath-IG (2.0) |
|---|---|---|---|---|
| VGG-16 | 42.3 | 44.7 | 45.0 | 47.2 |
| VGG-19 | 43.4 | 45.2 | 45.3 | 48.9 |
| ResNet-50 | 45.2 | 46.9 | 46.6 | 50.0 |

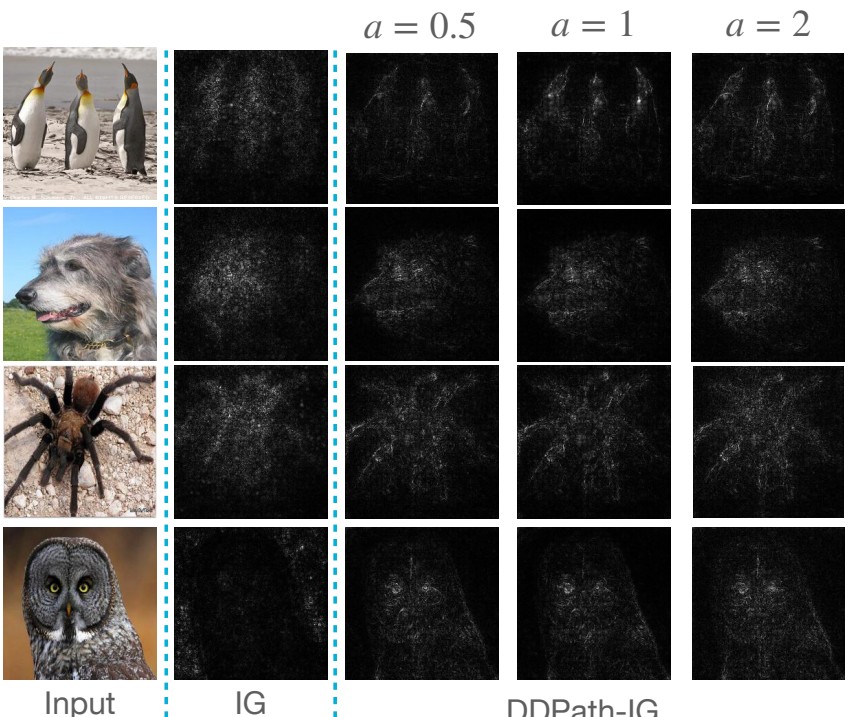

Figure 10: Saliency maps generated by different scaling schemes with $a \in \{0.5, 1, 2\}$.

