# OpenReview forum: "Denoising Diffusion Path: Attribution Noise Reduction with An Auxiliary Diffusion Model"
_NeurIPS.cc/2024/Conference — NeurIPS 2024 poster_

### Official Review · Reviewer_rjLS · 2024-07-03

**Soundness:** 3
**Presentation:** 2
**Contribution:** 3
**Rating:** 5
**Confidence:** 2

**Summary:**

This paper proposes an approach to interpreting deep neural networks, specifically they focus on a path-based approach where they utilize the denoising ability of the diffusion model to construct a path to be integrated to compute the attribution of the target image.

**Strengths:**

1. I am not an expert in the field of interpretability, and the idea of combining diffusion modeling with a path-based approach seems novel to me.

2. Better quantitative results were achieved compared to existing path-based methods

**Weaknesses:**

1. The presentation in Section 3 is somewhat abbreviated, for example, there is no definition of the lower x^', which can be confusing to those unfamiliar with this area.

2. The proposed method requires the introduction of classifier-guided diffusion, which requires a pre-trained diffusion and noise-aware classifier, and it is not clear whether the method is applicable when not belonging to a predefined category or when the category is difficult to define.

**Questions:**

see weaknesss

**Limitations:**

The authors briefly discuss their limitations

---

> ### Author Rebuttal · Authors · 2024-08-06
>
> Thank you for appreciating the novelty of the proposed method and better quantitative results compared to existing methods. We would like to address your concerns as follows:
>
> **Weakness 1: Presentation in Section 3**
>
> Thank you for pointing out these issues. We will provide a more detailed description of integrated gradient (IG) in Section 3 so that it will be more friendly to those readers who are unfamiliar with this area. Specifically, the $x’$ denotes a baseline image, *i.e.*, starting point of the integration path, which can be a **black** image or a **noisy** image.
>
> **Weakness 2: Lack of introduction of classifier-guided diffusion and whether the method is applicable when no predefined category is given**
>
> Thank you for the suggestion. In the current version of the manuscript, we have mentioned the classifier-guided diffusion model [1] in Section 2. To enhance the readability, we will add a brief introduction of the classifier-guided diffusion model in Section 3 in the future version as below: “The classifier-guided reverse process conditioned the $p_{\theta} (x_{t}|x_{t+1})$ on class label $y$, then deriving a sampling strategy of $x_{t-1} \sim \mathcal{N} (\mu + s \Sigma \nabla_{x_{t}} \log p_{\phi}(y|x), \Sigma)$. In this paper, we let the $\mu = \mu_{\theta} (x)$ to ensure the intermediate images are centered at the target image.”
>
> For the second question, when there are no pre-defined categories for target images, the DDPath can also work because the pre-trained diffusion models and classifiers are fixed, and any target image can obtain classification results from the final Softmax output and corresponding saliency maps through back-propagation. However, the only concern is that we cannot guarantee the correctness of the classification result and whether this classification result matches well with the saliency map.
>
> [1] Diffusion Models Beat GANs on Image Synthesis

---

> > ### Author Response · Authors · 2024-08-13
> > **Response to Reviewer rjLS**
> >
> > Dear reviewer rjLS,
> >
> >     We sincerely appreciate your time and valuable feedback. We would be grateful to know if our response is satisfactory or if there are any additional points we should address. Looking forward to hear from you.

---

### Official Review · Reviewer_2oA3 · 2024-07-04

**Soundness:** 3
**Presentation:** 3
**Contribution:** 3
**Rating:** 6
**Confidence:** 3

**Summary:**

This paper proposes a Denoising Diffusion Path (DDPath) to address the challenge in path-based attribution methods, where intermediate steps often deviate from the training data distribution. By leveraging the denoising power of classifier-guided diffusion models, DDPath constructs a piece-wise linear path to ensure gradual denoising, resulting in cleaner and more interpretable attributions. It also demonstrates that DDPath satisfies the axiomatic properties and can be integrated seamlessly with existing methods like Integrated Gradient.

**Strengths:**

* The proposed DDPath is interesting as it intuitively addresses the challenges in the previous works.
* The paper validates the effectiveness of DDPath through comprehensive experiments on various interpretation methods and classification models, along with ablation studies.

**Weaknesses:**

* I am not sure about the details of the Algorithm 1. What is the input timestep for the pre-trained diffusion model and the classifier? Do noisy baselines $\textbf{x}_{t}$ align with the noise scheduling in diffusion models?

* I am confused about the use of the diffusion model. Why the diffusion model is used to estimate the noise from the target image $\textbf{x}$? Is the target image a noisy image?

**Questions:**

* Please clarify the details of the diffusion model mentioned in the weaknesses.

* Although DDPath requires multiple feed-forward of the diffusion model and the classifier, there is no analysis of the time-complexity comparison with other methods.

* What about using denoising task weights in DTR [1] instead of a linear scaling scheme? For example, defining each scaling factor as $\rho = 1 - {(\frac{t}{T})}^{\alpha}$ and  $\kappa = {(\frac{t}{T})}^{\alpha}$, and experiments with $\alpha = 0.5, 2$ provide a tighter connection to the diffusion model.

[1] Park et al., Denoising Task Routing for Diffusion Models, ICLR 2024.

**Limitations:**

The authors have discussed them.

---

> ### Author Rebuttal · Authors · 2024-08-06
>
> Thank you for appreciating the interesting idea of the proposed DDPath and comprehensive experiments. We would like to address your concerns as follows:
>
> **Weakness 1: About details of the Algorithm 1**
>
> First, the input timestep is $t \in [0, \ldots, T-1]$, and $t$ also denotes the steps along the path.
>
> Second, we would like to clarify that the $x’$ is the baseline, and the $x_{t}$ are sampled by the pre-trained diffusion model. Hence, the sampled images $x_{t}$ are aligned with the noise scheduling process, and they consist of the attribution path.
>
> **Weakness 2: About the diffusion model**
>
> We apologize for the confusion. Here, we clarify the motivation for using a diffusion model as follows. Intuitively, we intend to investigate the denoising power of diffusion models in reducing explanation noise. However, the unconditional diffusion models cannot guarantee that the finally sampled image is consistent with a target image to be explained, furthermore, unconditionally sampled images have NO class or semantic information. Consequently, we applied the classifier-guided diffusion model to generate step images within the attribution path.
> Specifically, the target image $x$ is an RGB image rather than a noisy image, distinct from the noisy images typically associated with diffusion models, as shown in Algorithm 1. Our implementation focuses on the Reverse process to sample step images, and as outlined in line 8 of Algorithm 1, the sampling mean incorporates class probability $p_{\phi}(y|x_{t})$, aligning our approach with the generative nature of diffusion model.
>
> **Question 1: Clarification on diffusion model**
>
> Please refer to the responses of Weakness 1 and Weakness 2 for clarification on the diffusion model.
>
> **Question 2: Time-complexity comparison**
>
> Thanks for pointing out this issue. For traditional attribution methods like IG, the time complexity is $\mathcal{O}(n)$, where $n$ denotes the number of forward-back processes with respect to the target model. When combined with DDPath, for example, the DDPath-IG, the time-complexity increases due to the reverse sampling process of classifier-guided diffusion models, i.e., $\mathcal{O}(m)$, where $m$ is the number of the guided reverse sampling process. Hence, within one attribution step, the time complexity of DDPath-IG is $\mathcal{O}(m) + \mathcal{O}(n)$. In Table R1, we compare the attribution time of IG and DDPath-IG, due to the simple sampling strategy we use, the DDPath-IG requires more time. However, the proposed DDPath has the potential to reduce sampling steps (or the path length) by combining more efficient diffusion sampling strategies with fewer sampling steps. Then, the DDPath will also be more efficient in practice. This is what we are working on for the next version of DDPath.
>
> Table R1 Comparison of attribution time for one image using VGG-19 as the target model
>
> | Method | Time (s) |
> | --- | --- |
> | IG | 2.5 |
> | DDPath-IG | 22.5 |
>
> **Question 3: About denoising task weights in DTR**
>
> Thanks for your valuable comment. The DTR is an interesting idea of creating separate information pathways within a single diffusion model by selectively activating different subsets of channels for each task. Per your concern, we evaluated our DDPath by setting $\rho = 1 - (\frac{t}{T})^{\alpha}$ and $\kappa = (\frac{t}{T})^{\alpha}$ with both $\alpha = 0.5$ and $\alpha = 2$; and we note that the linear scaling used in this paper is equal to that of $\alpha = 1$.
>
> As shown in Table R2, we can see that the DDPath-IG surpasses the baseline IG among different $\alpha$ values, indicating the effectiveness of our DDPath. When $\alpha=2$, the weight of the mean term decreases slowly at the early step, ensuring better preservation of the main object in the images. Besides, the weight of the class-related variance term increases fast at higher steps, enabling better preservation of discriminative information and object details, and this is consistent with the mechanism of task weights in [1]. In contrast, when $\alpha=0.5$, the variance weight increases fast at early steps while the noises are still severe, hence, the class-related information can be affected by the noises while influencing the classification results and attribution qualities. The setting of $\alpha=1$ is a trade-off in our experiments.
>
> For visualizations of different settings, we showcased the corresponding saliency maps in **Figure A3** in the attached **PDF** file. This experiment helps us complete the study of the scaling scheme.
>
> Table R2 Scaling with different $\alpha$ values using VGG-19 target model
>
> |  | IG | DDPath-IG (0.5) | DDPath-IG (1.0) | DDPath-IG (2.0) |
> | --- | --- | --- | --- | --- |
> | Insertion$\uparrow$ | 23.2 | 26.6 | 27.8 | 27.4 |
> | Deletion$\downarrow$ | 13.5 | 12.4 | 12.1 | 12.0 |
> | Overall$\uparrow$ | 9.7 | 14.2 | 15.7 | 15.4 |
>
> [1] Park et al., Denoising Task Routing for Diffusion Models, ICLR 2024.

---

> > ### Comment · Reviewer_2oA3 · 2024-08-08
> >
> > Thanks for the response. It well addressed my concerns and I appreciate the detailed answers have enhanced my understanding of the paper. I will increase my rating to weak accept.

---

> > > ### Author Response · Authors · 2024-08-08
> > > **Response to comment by Reviewer 2oA3**
> > >
> > > We sincerely appreciate your valuable insight and positive feedback. We will include the additional experiments and discussions in the next version.

---

### Official Review · Reviewer_qJDk · 2024-07-10

**Soundness:** 3
**Presentation:** 3
**Contribution:** 3
**Rating:** 7
**Confidence:** 3

**Summary:**

This paper proposes Denoising Diffusion Path (DDPath), a method to reduce noise in path-based attribution for deep neural networks. DDPath uses diffusion models to create a path with decreasing noise, resulting in clearer attributions. It can be integrated with existing methods like Integrated Gradients and maintains axiomatic properties. Experiments on ImageNet and MS COCO show DDPath outperforms traditional methods in producing clear explanations and improving metrics like Insertion and Deletion scores. The paper includes analysis, comparisons, and discussion of limitations.

**Strengths:**

- Smart problem framing: The paper identifies a key issue in attribution methods - noise accumulation. It cleverly uses diffusion models' denoising ability to address this, showing good understanding of both attribution techniques and recent advances in generative models.
- Clear method explanation: The proposed DDPath method and its variants are explained clearly, with the paper showing how it maintains important properties while improving interpretability. The mathematical exposition is very clear.
- Thorough testing: The experiments compare the method on ImageNet and COCO datasets using multiple metrics (Insertion, Deletion, AIC) against several baselines. Ablation studies provide insights into how different components affect performance.

**Weaknesses:**

- Computational cost: The method requires significantly more sampling steps (250) than traditional attribution methods, which could limit its practical application in time-sensitive scenarios or for large-scale analyses.
- Marginal quantitative improvements: While the paper shows improvements over baseline methods, the differences in quantitative metrics (e.g., Insertion and Deletion scores in Table 1) are relatively small. This raises questions about the practical significance of the improvements, especially given the increased computational cost.
- Unconvincing visual results: The saliency maps presented in Figure 3 do not demonstrate a dramatic improvement over previous methods. While there are some differences, they are subtle and it's not immediately clear that DDPath provides substantially clearer or more interpretable attributions than existing techniques.
- Limited to classification models: The paper only demonstrates the method's effectiveness on image classification tasks. There's no exploration or discussion of how DDPath might apply to or perform on non-classification models, such as regression, detection, or generative tasks.
Reliance on pre-trained diffusion models: The approach depends on having a suitable pre-trained diffusion model, which may not always be available for all domains or tasks.

**Questions:**

- Does image variation resulted from diffusion model demonstrate more correlation with image semantics than previous image variation methods?
- Does the improvement in attribution quality provide significant improvement in attribution-based applications?
- Does the attribution quality vary as pertained diffusion model size/quality varies?
- If an image is generated from adversarial attack on diffusion model, will the attribution quality be impacted?

**Limitations:**

The author discusses limitation of the method that it requires longer paths.

---

> ### Author Rebuttal · Authors · 2024-08-06
>
> Thank you for appreciating the smart problem framing, clear method explanation, and thorough testing. We would like to address your concerns as follows:
>
> **W 1: Computational cost**
>
> We acknowledge that the current DDPath might be computationally inefficient. However, we argue that the consistent quantitative performance gains can offset its increased computational cost. As demonstrated in Figure 4, DDPath exhibits superior denoising capabilities with more sampling steps, a characteristic NOT observed in other methods. Also, we agree with the reviewer that improving the computation efficiency is important in future studies, *e.g.*, combining Consistency models [ICML 2023] and one-step diffusion [CVPR 2024].
>
> In addition, we emphasize that this paper primarily explores the alternative path formulation based on advanced diffusion models, hoping it can inspire more exciting studies on diffusion-based DNN attribution.
>
> **W 2: Marginal quantitative improvements**
>
> First, while the DDPath did not achieve a substantial performance gain over existing SOTA methods, it consistently demonstrated competitive improvements. In previous studies, the Score-CAM (Ins. 38.6) exhibited a 2.9% increase in insertion AUC when compared to the baseline Grad-CAM (Ins. 35.7) [CVPR 2020]; the CGC [CVPR 2022] improved the Insertion score to 52.16 against the baseline with 48.60 (3.56% improvements). In our paper, the DDPath-IG surpassed the IG, BlurIG, and GIG by 4.2%, 3.1%, and 2.3%, respectively. Additionally, from cases in Figure 3, we can see the superiority of DDPath in terms of Insertion and Deletion curves, demonstrating that DDPath captures more discriminative class-aware information. Hence, these findings suggest that DDPath represents a promising potential for future advancements in DNN attribution.
>
> Second, this paper is the first exploration of integrating diffusion models into DNN attribution, which has shown considerable effectiveness, and practically, since the DDPath can be fused with forward and backward processes of DNNs, it potentially provides new perspectives for explainable training and trustworthy AI.
>
> **W 3: Unconvincing visual results**
>
> We clarify that we aim to mitigate the noise in saliency maps to ensure that the model's predictions are solely driven by relevant features. This is critical to XAI. In Figure 3, we can see less noise obtained by DDPath, especially in discriminative regions like the heads and wings of birds, or the legs of lobster. In Figures 6 and 7 in the Appendix, we can see more significant visual results, especially those images containing multiple objects.
>
> **W 4: Limited to classification models**
>
> **Application tasks**: Current popular studies in XAI, especially the path-based methods, focused on image classification task, so as to this paper. Theoretically, the DDPath does not support the regression or detection tasks due to the step images are generated by classifier-guided diffusion model without regression or detection guidance.
>
> **Reliance on pre-trained diffusion models**: Recall that our motivation is to make intermediate images along the path to correlate with the original data distribution and then reduce the prediction bias and explanation noise. Since the original diffusion model was trained with no conditions, we applied the classifier-guided diffusion model to make the intermediate images within the path have class information. Fortunately, the diffusion model in [1] was trained on the ImageNet dataset, facilitating our interpretation study on ImageNet. Indeed, for the interpretation of other domains, it is better to have a classifier-guided diffusion model pre-trained on those domains.
>
> **Q 1: Image variation and semantics**
>
> Taking integrated gradient (IG), a typical attribution method, as an example, the intermediate images are generated by independently injecting image intensities into a black baseline image. This generation process has no correlation with semantic information. For DDPath, in line 8 of Algorithm 1, the sampling mean involves the conditional probability $p_{\phi}(y | x_{t})$, which incorporates class information into generated intermediate images.
>
> **Q 2: Improvement in attribution-based applications**
>
> One of the typical attribution-based applications is weakly-supervised localization, i.e., localizing objects in the images with only class labels rather than bounding boxes. The pointing game in section 5.4 is also an application that verifies the performance of localizing objects through saliency maps, where the annotated bounding boxes in MS COCO are used as ground-truth labels. The results in Table 2 (of the paper) show improvements obtained by DDPath.
>
> **Q 3: Diffusion model size**
>
> To investigate the diffusion model size, we apply the released diffusion models by [1]. Note that these diffusion models of different sizes correspond to different image resolutions, including $64\times 64$, $128 \times 128$, $256 \times 256$, and $512 \times 512$. The visualization results can be found in Figure A1 in the attached PDF file. We can see that larger models generated larger resolution of saliency maps, and they illustrate more fine-grained details.
>
> **Q 4: Attribution for adversarial samples**
>
> Thanks for your concern. We applied two approaches to generate adversarial samples, one is the Fast Gradient Sign Attack (FGSM) described by Goodfellow et. al [2], and the other is adding simple Gaussian noise. We compared the results of IG and DDPath-IG in terms of Insertion and Deletion values, these results and the saliency maps are shown in Figure A2 in the attached PDF file. It is interesting that the IG generated saliency maps with degraded quality, while the DDPath-IG are more robust to adversarial samples (FGSM and Gaussian). Additionally, the Insertion and Deletion values of DDPath-IG are superior to IG.
>
> [1] Diffusion Models Beat GANs on Image Synthesis, NeurIPS 2021.
>
> [2] Explaining and Harnessing Adversarial Examples, ICLR 2015.

---

> > ### Comment · Reviewer_qJDk · 2024-08-10
> >
> > Thanks for your response. These addressed many of my concerns. I have one additional questions regrading to your response to W 4: Limited to classification models. Can this method be extended to study embedding models (such as CLIP) and attribute image features for downstream tasks?

---

> > > ### Author Response · Authors · 2024-08-11
> > > **Response to Concern on embedding models (such as CLIP)**
> > >
> > > Thanks for your valuable feedback.
> > >
> > > First, the proposed DDPath is model-agnostic and **CAN** be extended to embedding models such as CLIP. Specifically, one can fine-tune the CLIP image encoder coupled with a classifier (such as a Linear layer). Then, the DDPath can work on this trained model (CLIP image encoder + classifier) to obtain saliency maps using DDPath-IG. Theoretically, DDPath is adaptive to both ViT and CNN backbones pre-trained by CLIP. However, to the best of our knowledge, path-based attribution studies have not been focused on ViT models, and one of the important reasons is that using path-based attribution methods can result in grid-like artifacts caused by the image-patch nature of ViT. Also, this may provide us with an interesting further research direction in XAI.
> > >
> > > Second, we conducted additional experiments by simply fine-tuning a linear classifier with CLIP’s image encoder (ResNet-50) on CIFAR100 and Flowers102 datasets. We report the top-1 accuracy and interpretability metrics. Note that the primary goal of this experiment is not to achieve optimal classification accuracy but to demonstrate the feasibility and effectiveness of the DDPath method. We can see that DDPath-IG outperforms IG on both Insertion and Deletion values, demonstrating the effectiveness of applying the DDPath to downstream tasks using pre-trained embedding models.
> > >
> > > Table R1 Classification accuracy (%) and interpretability metrics (IG / DDPath-IG) obtained by fine-tuning CLIP-ResNet-50
> > >
> > > |  | CIFAR100 | Flowers102 |
> > > | --- | --- | --- |
> > > | Accuracy | 72.55 | 88.25 |
> > > | Insertion$\uparrow$ | 25.6 / **29.2** | 23.3 / **26.7** |
> > > | Deletion$\downarrow$ | 14.9 / **12.8** | 17.5 / **16.2** |
> > >
> > > Through the above discussions and experiments, we hope we have solved your additional concern.

---

> > > > ### Author Response · Authors · 2024-08-13
> > > > **Response to further concerns by Reviewer qJDk**
> > > >
> > > > Dear reviewer qJDk, we would like to express our sincere gratitude once again for your response and insightful questions. We are eager to know if our further explanations and experimental results can well addressed your concerns.

---

> > > > > ### Comment · Reviewer_qJDk · 2024-08-14
> > > > >
> > > > > Thank you for your response. I have raised my score.

---

### Author Rebuttal · Authors · 2024-08-06

Dear Reviewers,

We thank all the reviewers for their thorough summaries and valuable feedback. All the reviewers appreciated the **Novelty** and interesting idea of this work. The reviewers appreciate that our DDPath, combining diffusion models and DNN’s attribution, is Smart, Interesting, and Novel (**qJDk**, **2oA3**, **rjLS**) while intuitively addressing the challenges in previous works (**2oA3**), the experiments are comprehensive (**qJDk**, **2oA3**) and demonstrate the effectiveness of DDPath and better quantitative results (**qJDk**, **2oA3, rjLS**), clear method explanation and mathematical exposition (**qJDk**).

We have posted detailed responses to each reviewer and deeply appreciate your further feedback on whether our responses adequately address your concerns. If you have any additional comments or questions, we will try our best to address them.

Per Weakness 3 and Questions 3 and 4 of **qJDk**, Question 3 of **2oA3**, the following **attached pdf file** provide more saliency maps generated by diffusion models of different sizes (**qJDk**), adversarial examples (**qJDk**), and different scaling schemes (**2oA3**):

- Figure A1: Saliency maps generated by DDPat-IG using diffusion models of varying sizes.
- Figure A2: Saliency maps for adversarial examples generated by FGSM and Gaussian.
- Figure A3: Saliency maps generated by different scaling schemes with $\alpha \in \{0.5, 1, 2\}$.

Best,

The authors

---

### Decision · Program_Chairs · 2024-09-25

**Decision:**

Accept (poster)

**Comment:**

This paper has received consistent feedback from all three reviewers. The reviewers engaged in thorough discussion and rebuttal, with most of their concerns being addressed, ultimately reaching a consensus. The paper proposes a novel Denoising Diffusion Path (DDPath) for DNN attribution, which can be easily implemented using a pre-trained classifier-guided diffusion model. Comprehensive experiments  and ablation studies validate the effectiveness of DDPath. Therefore,  the AC  has decided to accept this paper.